# The Influence of Connectedness to Nature on Psychological Well-Being: Evidence from the Randomized Controlled Trial *Play&Grow*

**Tanja Sobko** [1,*] and **Gavin T. L. Brown** [2]

1   Faculty of Science, University of Hong Kong, Hong Kong 852, China
2   Faculty of Education & Social Work, University of Auckland, Auckland 1142, New Zealand; gt.brown@auckland.ac.nz
*   Correspondence: tsobko@hku.hk; Tel.: +852-2299-0611; Fax: +852-2559-9114

**Abstract:** Urbanized children today have fewer opportunities to interact with nature which may lead to a greater risk of mental health problems. The objective of this randomized controlled trial was to investigate which particular changes in connectedness to nature (CN) would improve psychological well-being (PW) in young children. Six hundred and thirty-nine preschoolers (52.0% boys, age 34.9 ± 9.5 months) participated in *Play&Grow*, an early environmental education intervention. Children's CN and PW were evaluated by parents before and after the program with validated measures; the CNI-PPC (four factors) and the SDQ, Strength and Difficulties questionnaire (five factors), respectively. The effectiveness of the intervention on the primary outcomes (CN, PW) as well as the relationship between them was analyzed in a repeated measures path model with intervention status as a causal predictor. Specific CN factors consistently increased ProSocial behavior and reduced Hyperactivity and Emotional problems. In summary, this study showed that the previously reported impact shifted from the total CN score to the specific CN factors. The *Play&Grow* intervention positively increased children's CN and improved some aspects of psychological well-being in children which is a preliminary evidence of developmental benefits of connecting young children with nature. Our results indicate promising direction of action for the improvement of families' psychological health.

**Keywords:** connectedness to nature; outdoor time; mental health; psychological development; psychological well-being; preschooler; sense of responsibility for nature; sense of enjoyment of nature; strength and difficulties questionnaire (SDQ); pro-social behavior

## 1. Introduction

The mental health of urbanized children is of increasing concern. It is estimated that 17.6% of preschoolers have some form of mental disorder [1]. Certain behavioral problems in children are likewise becoming a prominent issue, particularly in big cosmopolitan cities like Hong Kong [2]. Reduced parental perceptions of psychological problems were found for children who were rated higher for responsibility for, enjoyment of, and awareness of nature [3]. However, that study did not reveal whether participation in nature had any impact on those relations. This study explicitly examines changes in how nature relatedness connects to psychological well-being after a randomized control trial that increased connectedness to nature.

It has been reported that experiencing nature-based activities improved not only CN but also happiness, mediated by better emotional regulation [4], and time spent in nature can increase children's enjoyment of nature and environmental sensitivity [5]. In addition to emotional benefits, being in nature may positively contribute to a psychosocial development. Prosocial behavior, the voluntary and intentional behavior that benefits others [6], is claimed to have a positive effect on the development of sociability at a young age and is

therefore a focus of many recent studies. Prosocial behavior, with its roots in toddlerhood, develops with age from the simple forms of instrumental helping to more sophisticated types of sharing and comforting [7,8]. Two precursors of prosocial behaviors [9] are (a) empathy, when experience of other's pain arises [6], and (b) responsibility with a feeling of guilt if an individual perceives oneself as being responsible, but unable to help [10].

It has not been yet investigated if the sense of responsibility, verified to be a significant motivator for prosocial behaviors [11–13], could be transferred from being responsible for nature to the responsibility felt towards other people. As for nature interventions, theoretically, perceiving natural surroundings and taking care of plants or animals should enhance children's cognitive capabilities, including perception, sustained attention, and control, which are fundamental elements for advanced social functions. Time in nature becomes therefore crucial as it is linked with increased pro-environmental behaviors later in life [14].

Finally, our group has recently shown in a robust RCT that indoor and outdoor nature-related activities not only improved CN but also brought about positive changes in gut microbiota in pre-school children, which may in turn lead to healthier lifestyles with better psychological management and behaviors [15]. Following the intervention, fecal serotonin level and gut microbiota profiles were measured by ELISA and 16S rDNA amplicon sequencing, respectively. The intervention improved children's connectedness to nature, particularly, their responsibility towards nature. The gut microbiota of children was improved by stabilizing the abundance of Roseburia, as well as related fecal-serotonin levels. Moreover, children were overall less stressed, and, in particular, less often angry. That work demonstrated for the first time the impact of nature-related activities on gut microbiota and fecal serotonin, and their relationship in the improvement of psychosocial behavior of preschool children. The *Play&Grow* early environmental education program, with its unique Connectedness to Nature component, is designed to increase biophilia and positive health outcomes for preschoolers. This intervention allows interaction with the natural outdoor world and has proven to be effective in encouraging healthy lifestyle behaviors in families with preschoolers in prior experiments [16–18]. The main objective of this sub-study was to investigate a potential association between the CN and the psychological well-being of two-to-five-year-old children following the *Play&Grow* intervention, reflected in emotional and behavioral problems before and after the intervention.

## 2. Results

The previously reported study [3] validated the newly created CN scale by showing that it suppressed negative aspects of psychological well-being measured by the Strengths and Difficulties Questionnaire (SDQ, [19]) used to evaluate participants' psychological development in the past three months; and increased its prosocial aspects, without distinguishing between those in or outside the RCT. This study replicates and extends that result by including repeated measures for both inventories and accounting for the impact of the RCT intervention.

The inter-time correlation for the SDQ construct was also moderate ($r = 0.72$) and the residual correlations for scale score repetitions were moderate ($0.40 < r < 0.60$). The average correlation for CN time 1 to time 2 was $r = 0.36$ with $SD = 0.15$; the average for SDQ time 1 to time 2 was $r = 0.50$ with $SD = 0.06$. A test of difference between those two mean values of $r$ with $N = 639$ has $z = |3.09|$, $p_{(2\text{-tail})} = 0.002$; therefore, it is possible to conclude that the consistency over time is greater in the SDQ than the CN.

More rigorously, we inspected the 95%CI for an overlap between IG and CG conditions for covariances with a <50% proportion of overlap in weighted average confidence intervals as the basis for determining if observed values differed by more than chance [20]. The length of CI arm was $2 \times$ se weighted by group size, with a proportion of overlap (POL) determined by the distance from the Upper Limit of the lower value to the observed statistic of the higher value divided by the weighted CI arm. Values under 50% were deemed to be statistically significant [20]. Three covariances from Time 1 to Time 2 variables had POL

<50% between IG and CG, two of which were within CN and only one in SDQ. In these three cases, the standardized covariance (i.e., correlation) was smaller in the IG than the CG. Specifically, for Responsibility ($r_{EG}$ = 0.27, $r_{CG}$ = 0.38), Empathy ($r_{EG}$ = 0.54, $r_{CG}$ = 0.65), and for Peer Difficulties ($r_{EG}$ = 0.51, $r_{CG}$ = 0.65). Together, these indicate that there was less stability in the parental scoring of connectedness to nature after the intervention and more consistency in scoring of psychological well-being. This suggests the *Play&Grow* intervention disrupted the status quo for the CN more than the SDQ.

### 2.1. Relationship between the CN and the SDQ across Time after the Intervention

The previously reported model of CN to SDQ [3] was used as the basis for this study. The SDQ total score was removed because it was linearly dependent as a sum of 4 'difficulty' factors (hyperactivity, peer, conduct, and emotionality). The current approach allows detailed examination of how CN scales relate to the five different facets of SDQ (1 'strength' and 4 'difficulty' factors) post-intervention; this prevents loss of signal if only total score were used. An advantage of structural equation modeling with sufficiently large samples is that it can provide analysis of this type of modeling easily.

The prosocial scale was consistently inverse but weakly loaded on the SDQ scale (β = −0.33). Before the intervention (Figure 1), child Prosocial behavior ratings were positively predicted by responsibility for and empathy for nature (β = 0.10; 0.25, respectively). In contrast, hyperactivity was suppressed by responsibility for nature (β = −0.06) and peer problems were suppressed by empathy for nature (β = −0.13). Overall, greater CN at prior to the intervention suppressed the total SDQ at the same time 1 (β = −0.21).

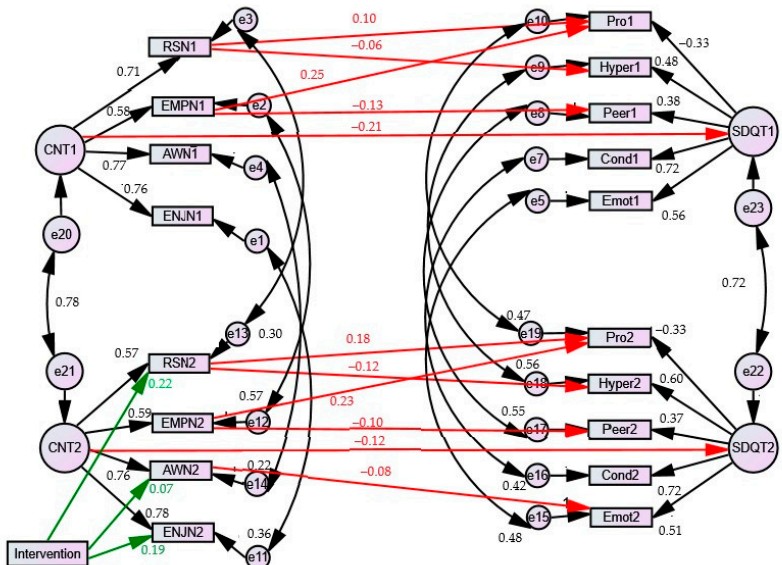

**Figure 1.** CNI as predictor of SDQ score. Green when positive; R2 values for SDQ scores displayed. The fit of this model is good. $\chi^2$ = 331.64, *df* = 128, $\chi^2/df$ = 2.59, *p* = 0.11; CFI = 0.95; gamma hat = 0.97; RMSEA = 0.051 (90% CI = 0.044–0.058, $p_{close}$ = 0.39); SRMR = 0.052. Intervention Status = Intervention as a predictor variable; Pro1, Pro2 = Before/After intervention score of children's ProSocial behaviors; Hyper1, Hyper2 = Before/After intervention score of children's Hyperactivity; Peer1, Peer2 = Before/After intervention score of children's Peer problems; Cond1, Cond2 = Before/After intervention score of children's Conduct Problems; Emot1, Emot2 = Before/After intervention score of children's Emotional problems; EMPN1, EMP2 = Before/After intervention score of Empathy for nature; RSN1, RSN2 = Before/After intervention score of Responsibility towards nature; ENJN1, ENJN2 = Before/After intervention score of Enjoyment of nature; AWN1, AWN2 = Before/After intervention score of Awareness of nature.

The intervention positively loaded onto responsibility for nature (β = 0.22), awareness of nature (β = 0.07), and enjoyment of nature (β = 0.19). Consequently, the effect of

responsibility for nature increased for prosocial strengths (β = 0.18) and more strongly suppressed hyperactivity (β = −0.12). Empathy for nature retained the same paths with nearly the same values for increasing prosocial strengths and suppressing peer problems (β = 0.23; −0.10, respectively). A new relationship was seen in greater awareness of nature suppressing emotional difficulties (β = −0.08). The overall effect of CN on SDQ at time 2 was smaller than time 1 (β = −0.12), suggesting that specific scales had picked up unique aspects of the shared variance.

When we ran a multigroup-nested invariance test between Intervention and Control groups without the intervention variable, the change in CFI for metric equivalence was =0.000; the change to scalar equivalence was =0.016. Together, these mean that the regression weights were equivalent between conditions, but the intercepts differed by more than chance. In a repeated measures study, McArdle [21] argues that this is sufficient for longitudinal data analysis, and even though the slope values of factors to items are equivalent between groups, the auto-regressive Time 1 to 2 covariances and the intercepts of scale scores to the CN factor and/or the SDQ factor differed by more than chance between groups.

Examination of the total effect of each CNI factors on the SDQ subscales showed an overall tendency that higher scores of the CNI were associated with fewer problems in the SDQ (Table 1). The direct and indirect effects of the intervention as a dummy variable (0 = control, 1 = intervention) on the three CN factors are shown in Figure 1. The effect of the Intervention on CN is communicated to the SDQ scales as per the regression paths shown in Figure 1. For example, the increase in ProSocial SDQ is achieved by the intervention effect on Responsibility; the multiplicative effect of 0.22 × 18 produces an indirect effect of 0.032 on the SDQ ProSocial at Time 2. When converted to an effect size $f^2$ this is 0.032 (small) [22]. Hence, the *Play&Grow* intervention had a notable effect on three CN scales, and consequently small indirect effects on three of the SDQ scales, most notably Prosocial strengths.

**Table 1.** Effect sizes of Intervention.

| Dependent Variable | Beta Values | | Total $R^2$ | $f^2$ |
| --- | --- | --- | --- | --- |
| | Direct | Indirect | | |
| Connectedness to Nature | | | | |
| Responsibility | 0.387 | | 0.387 | 0.63 |
| Awareness | 0.098 | | 0.098 | 0.11 |
| Enjoyment | 0.285 | | 0.285 | 0.40 |
| Strengths & Difficulties | | | | |
| ProSocial | | 0.032 | 0.032 | 0.03 |
| Hyperactivity | | −0.022 | −0.022 | −0.02 |
| Emotionality | | −0.004 | −0.004 | 0.00 |

Note. Responsibility = Responsibility towards nature, ProSocial = ProSocial behaviors, Emotionality = Emotional problems, Enjoyment = Enjoyment of nature, Awareness = Awareness of nature.

## 2.2. Intervention Group (IG) vs Control Group (IG)

Having established that the relationship of CN to SDQ was affected by the intervention, it is useful to examine the scale score differences between the IG and CG. The measurement models for CN and SDQ were invariant between the two groups but the structural model (Figure 1) only had metric equivalence (meaning item regressions were equivalent between groups). Scalar equivalence, meaning the start values for each factor to item regression, differed by more than chance (i.e., ΔCFI = 0.016). The fit of this model after fixing the metric weights but allowing intercepts to vary by group is good and notably better than the model in Figure 1 ($\chi^2$ = 469.39, *df* = 240, $\chi^2/df$ = 1.96, *p* = 0.16; CFI = 0.95; gamma hat = 0.98 RMSEA = 0.040 (90% CI = 0.034–0.045, $p_{close}$ = 1.00); SRMR = 0.051).

To examine the nature of the differences in starting values or intercepts, Table 2 presents intercept values and standard errors by group. The size of intercept differences was calculated as a proportion of the weighted 95% CI ($N \times 2 \times$ se). If this value >1.0, then the reported values do not reliably overlap. Differences in favor of the IG are marked in green, while those in favor of the CG are marked in red. Before the intervention, the CG had higher start values for three of the four CN scales and one of the SDQ scales. This means that any shift to higher values for the IG will have had to potentially overcome the original effect of being lower.

**Table 2.** Intercept values for CN and SDQ factors.

| Factor | IG | | CG | | Difference | | |
|---|---|---|---|---|---|---|---|
| | Intercept | se | Intercept | se | Diff | Weighted se | Diff/se |
| Connectedness to Nature (CN) | | | | | | | |
| Awareness1 | 3.986 | 0.029 | 4.081 | 0.047 | −0.10 | 0.07 | −1.43 |
| Responsibility1 | 3.429 | 0.032 | 3.552 | 0.053 | −0.12 | 0.07 | −1.66 |
| Empathy1 | 3.564 | 0.035 | 3.726 | 0.066 | −0.16 | 0.08 | −1.91 |
| Enjoyment1 | 3.844 | 0.031 | 3.83 | 0.052 | 0.01 | 0.07 | 0.19 |
| Awareness2 | 4.314 | 0.027 | 4.257 | 0.048 | 0.06 | 0.06 | 0.89 |
| Responsibility2 | 3.784 | 0.033 | 3.452 | 0.065 | 0.33 | 0.08 | 4.09 |
| Empathy2 | 3.781 | 0.034 | 3.846 | 0.063 | −0.06 | 0.08 | −0.80 |
| Enjoyment2 | 4.054 | 0.029 | 3.782 | 0.056 | 0.27 | 0.07 | 3.84 |
| Strengths & Difficulties Questionnaire (SDQ) | | | | | | | |
| ProSocial1 | 0.624 | 0.088 | 0.594 | 0.176 | 0.03 | 0.22 | 0.14 |
| Hyperactivity1 | 1.032 | 0.074 | 0.945 | 0.145 | 0.09 | 0.18 | 0.48 |
| Peer1 | 0.745 | 0.069 | 0.93 | 0.12 | −0.19 | 0.16 | −1.14 |
| Conduct1 | 0.569 | 0.015 | 0.589 | 0.029 | −0.02 | 0.04 | −0.55 |
| Emotional1 | 0.377 | 0.016 | 0.394 | 0.026 | −0.02 | 0.04 | −0.46 |
| ProSocial2 | 0.563 | 0.088 | 0.489 | 0.152 | 0.07 | 0.21 | 0.36 |
| Hyperactivity2 | 1.144 | 0.072 | 1.027 | 0.111 | 0.12 | 0.16 | 0.72 |
| Peer2 | 0.756 | 0.068 | 0.745 | 0.124 | 0.01 | 0.16 | 0.07 |
| Conduct2 | 0.569 | 0.014 | 0.618 | 0.026 | −0.05 | 0.03 | −1.45 |
| Emotional2 | 0.558 | 0.091 | 0.615 | 0.179 | −0.06 | 0.22 | −0.25 |

Note. Green means Intervention is higher than Control; Red = Control is higher than Intervention; ProSocial 1, ProSocial 2 = Before/After intervention score of children's ProSocial behaviors, Hyperactivity1, Hyperactivity2 = Before/After intervention score of children's Hyperactivity, Peer1, Peer2 = Before/After intervention score of children's Peer problems, Conduct1, Conduct2 = Before/After intervention score of children's Conduct Problems, Emotional1, Emotional2 = Before/After intervention score of children's Emotional problems, Empathy1, Empathy2 = Before/After intervention score of Empathy for nature, Responsibility1, Responsibility2 = Before/After intervention score of Responsibility towards nature, Enjoyment1, Enjoyment2 = Before/After intervention score of Enjoyment of nature, Awareness1, Awareness2 = Before/After intervention score of Awareness of nature.

After the intervention, two of the CN scales (i.e., Responsibility and Enjoyment) had higher intercepts for the IG group, while the CG had a higher intercept for one SDQ scale. These same two scales had POL <0% because the 95% CI did not overlap at all. In contrast, the proportion of overlap was >50% between EG and CG for all CN to SDQ regression weights. This indicates that the *Play&Grow* intervention shifted the start values of two CN scales upward, but did not change how strongly the CN factors influenced the SDQ scale scores.

To summarize, within the CN, the CG intercepts were higher at the beginning of the study for three of the factors, but by the end of the study, the IG was higher for two scales. Within the SDQ, the CG had higher start values for Peer problems and for Conduct problems at the end of the study. This pattern points to a positive impact of the intervention on 'Responsibility' and 'Enjoyment', while the children in the CG, who did not get this intervention, had parents who perceived greater conduct problems by the end of the study.

### 3. Discussion

To investigate the impact of Connectedness to Nature (CN), we conducted a randomized, controlled trial and studied its impact on changes on psychological well-being of preschoolers. A novel element, Connectedness to Nature, was integrated into the family-based *Play&Grow* intervention and discovered a range of positive outcomes related to healthy lifestyle for the Intervention Group. Unlike previous a cross-sectional study [3], the current study's design with 'before' and 'after' the intervention time points provides insights into some mechanism of how connectedness to nature contributes to children's psychological well-being. With regard to prosocial behaviors and hyperactivity, the responsibility toward nature (RN) was identified to be a strong beneficial factor. Children with higher RN showed lower levels of problems in prosociality and hyperactivity, and those who improved in their RN scores most exhibited greater increase in prosociality and decrease in hyperactivity. In addition to RN, empathy for nature was another positive and relatively strong predictor of prosocial behaviors. Moreover, for peer problems, awareness of nature was a significant protective factor. None of the CN factors showed direct or covariant association with conduct problems. The impact of the CN intervention on parental perceptions of strengths and difficulties warrants some specific consideration.

The positive changes in the Prosocial scores of participants in the IG after the *Play&Grow* intervention are in agreement with the recently published study measuring gut microbiome [15], where the *Play&Grow* program significantly improved the overall perceived stress, particularly anger frequency and prosocial behavior of children, and suggested a mechanistic link between these behaviors and the bacterial load when children are exposed to natural environments. Likewise, beneficial impacts of exposure to the natural environment on negative emotions (i.e., anger and sadness) was also highlighted in a systematic review [23]. Responsibility for Nature (RN) demonstrated not only a positive cross-sectional correlation but also a covariant relationship with prosocial behaviors in this study. This gives empirical support for the idea that when being responsible for nature in general, children tend to build responsibility in general including toward their peers, which would enhance their prosociality.

In addition to RN, children's prosocial behaviors were also positively linked to their levels of empathy for nature. The positive and constant association between empathy and prosocial behaviors provided empirical support for a generalized theory of empathy [6,9] that contributes to further improvement of children's prosociality. Overall, our results support some recent research which examined the potential benefits of a 4-h nature experience on children's mood, prosociality, and attitudes toward nature, suggesting a causal link between nature exposure and prosociality [24].

In this study, children's hyperactivity (one of the 'difficulties') was negatively related to RN, and when the RN improved, so was hyperactivity suppressed. These results implicate a connection between nature and cognitive benefits. It could be inferred that when in an unpredicted and spontaneous natural environment, children had a chance to train their attention through patience and concentration. Furthermore, children may have a chance to reflect on the consequences of their actions on the surrounding environment and their peers, which in turn led to more controlled impulsiveness. It is noteworthy that some research which is based on children's growth curves, showed a positive relation between outdoor hours and children's digit span scores, and an inverse relation between outdoor hours and inattention hyperactivity symptoms [25]. Some neural mechanisms, such as serotonergic system, may be behind the influence of RN on hyperactivity [26], but this hypothesis must still be proven empirically.

Our results indicated that children's emotional improvement levels were positively linked to an increasing awareness of nature. The exact mechanisms of how nature would help people feel more balanced and calmer is yet to be identified. The Enjoyment of Nature factor was covariant with awareness of nature, suggesting children who enjoyed nature paid more attention to and consequently had fewer emotional difficulties. This is consistent with previous empirical research showing that the participants with higher

CN reported more positive affect towards nature compared to those with lower CN, who were less used to spending time in nature [27]. Thus, we suggest that improvement in awareness of nature concomitant with greater enjoyment of nature might be beneficial for the emotional improvements via indirect paths involving other CN factors. At the same time, the relationships demonstrated here may not be unique to nature-related enjoyment, empathy, responsibility, and awareness. That is, perhaps enjoyment of any particular activity, empathy toward any vulnerable other, responsibility toward any aspect of one's life, and awareness of any ongoing activity would be predictive of positive psychological health. Future research could examine the specific relationship between nature-related enjoyment, empathy, etc., and psychological health, controlling for non-nature-related similar emotions.

Childhood well-being is also crucial for longer-term mental development [28,29]. The recent and ongoing novel coronavirus 2019 (COVID-19) outbreak has worsened the situation while imposing significant dramatically negative changes for the entire world's population: recent studies highlighted the issues of social isolation measures and their severe implications, such as family stress and continuous exposure of children to a significant risk of their well-being [30–32]. A recent study analyzing the outcomes of the home confinement implemented nationwide in China in response to COVID-19 pandemic highlighted that almost 70 percent of the children had increased daily screen time, and over 80 percent of the children had less than 2 h of daily outdoor activity, with one fifth reported as 'none' in the youngest children. The same study measured emotional and behavioral problems with the parent-rated Strengths and Difficulties Questionnaire (SDQ) and reported children experiencing problems in regard to their psychological well-being, and the youngest child students were more vulnerable than the older ones (19.3%, 16.7%, and 13.7% for low, middle, and high grades, respectively) [32]. This is a significant concern as exposure to natural outdoor spaces not only enhances children's physical well-being and their cognition but also improves their mental well-being [33–36]. The positive correlation between connectedness to nature (CN) and psychological well-being has been demonstrated in empirical studies on children, adolescents, and younger children [3,36,37]. Interestingly, children living in suburban areas had better mental development and academic achievements than their counterparts living in large urban and rural areas [38,39].

*Strengths and Limitations*

This study identified the relationships of children's emotional and behavioral difficulties with connectedness to nature using a multidimensional four-factor scale, instead of considering the unidimensional CN as a whole scale. As an intervention study with before and after measurements, it was possible to directly examine how changes in the CN variables influenced mental health outcomes. The younger age of the participants provided preliminary evidence to the design and effectiveness of a simple evidence-based nature intervention for preschoolers. However, some limitations of the study are acknowledged and the results of the effectiveness of the program can only be seen as indicative. Firstly, since the study was performed only in Hong Kong—although in a robust RCT format—the causal relationships observed here could vary in different cultures and societies. Secondly, while our results are very promising, we have not had an opportunity to investigate whether these positive changes have been sustained over the longer-term. It is recognized that to track any long-term changes a follow-up assessment is necessary and is therefore planned. Finally, the current study only examined the relationship of two measurements, the CNI-PPC and the SDQ. Future research needs to probe more the investigation of physical and psychological mechanisms, using other validated tools.

## 4. Materials and Methods

### 4.1. Study Design and Procedures

The current study was an intervention with pre- and post-intervention assessments. Participants were prospectively assigned to the *Play&Grow* intervention according to a

published protocol to evaluate the program's effect on health-related outcomes [19]. A locally developed, family-based program, *Play&Grow*, was tested in a two-arm, randomized controlled trial design, with masked outcome assessment at the University of Hong Kong. The RCT was conducted throughout 2017, and included staff training, Intervention sessions, and data collection from both the Intervention Group (IG) and the Control Group (CG). The G*Power 3.1 software was used to determine power [40]. In a two-group study of mean scores, and equal sized groups, a small effect size is Cohen's $d = 0.20$ and a medium effect size is $d = 0.50$ [22]. The conventional power threshold is $1-\beta = 0.80$. When each group has 150 people, the obtained power is $1-\beta = 0.99$ for $d = 0.50$; the power for the current study ($n = 639$ EG, 167 CG) is $d = 0.25$, $1-\beta = 0.82$. This indicates the study has sufficient power to detect reasonably small changes in means.

The rationale and earlier results are described in greater detail in the protocol and other publications [3,15,17,18]. *Play&Grow* is a structured intervention aimed at improving urban preschoolers' connectedness to natural environments through interactive games for children and providing guidance for parents/caregivers on taking care of their children. The program consists of 10 weekly sessions (Table 3). Each session lasted about 45 min and included 3 sections: "Active Play", "Food Fun", and "Nature Fun". In "Active Play" children are asked to mimic certain activities of animals (e.g., walking sideways like a crab). This is a warm-up activity. In "Food Fun", with the help of their parent/caregiver, children were guided to create edible figures using vegetables, for example, a car made out of a cucumber. They were encouraged to taste these vegetables as well. In "Nature Fun", children were invited to have fun with natural materials, such as making a smiling face using leaves, flowers, and small stones. Parents/caregivers were asked to fill two questionnaires about children's physical and psychological development before and after intervention. Parental consent was acquired prior to children's participation in the program.

**Table 3.** Details of the *Play&Grow* intervention, the detailed description of the relevant theories can be seen in [3,18]

| Session | Activities during the Program |
| --- | --- |
| 1 | **Introduction of environmental education program:** Setting goals, communication in the family. Introduce basic concepts regarding environmental awareness, parental feeding styles, and how these might relate to beliefs about healthy lifestyle and safety of children. Connectedness to Nature/outdoor play: active nature games, discovering nature, practicing awareness to sounds, touch, smells, temperature, etc. *Theories used:* The Social Ecological Model (SEM), used in the micro-level environmental settings to improve eating habits through exo-environmental linkages, such as greater caregiver environmental awareness, in all sessions. Self Determination Theory (SDT) and Erikson's Theory of Human Development (THD): on motivation and autonomy to set up goals in the family. |
| 2 | **Environmental education and Healthy eating:** Food and nature. Where the food comes from. Food groups and reading food labels. How much to eat? Develop caregivers' understanding regarding basic nutrition principles. Connectedness to Nature/outdoor play: active nature games, discovering nature, practicing awareness to sounds, touch, smells, temperature, etc. *Theories used:* SEM, Social Cognitive Theory (SCT) and Patterson's Social Interaction Learning Theory (SILT) to purposefully play with a planted vegetable and learn the relationship between food and nature, as well as to discuss basic nutrition principles with peer families in a group format. |

**Table 3.** *Cont.*

| Session | Activities during the Program |
|---|---|
| 3 | **Environmental education and Active play:** Methods to encourage active play in nature. Decrease inactive time. Motor skill development for children—the foundation. For an active life and safety. Develop of themes/skills regarding: moving for health caregivers and sedentary behaviors in families. Connectedness to Nature/outdoor play: active nature games, discovering nature, practicing awareness to sounds, touch, smells, temperature, etc. *Theories used:* SEM, SCT and SILT to develop games that encourage active play and interaction with peers and caregivers in nature. When cooperating with other families, participants were able to understand the importance of activity more efficiently and be more motivated. |
| 4 | **Environmental education and Sleeping time:** Develop caregivers understanding how being in nature influence sleeping behaviors, sleeping friend and sleeping routines. Connectedness to Nature/outdoor play: active nature games, discovering nature, practicing awareness to sounds, touch, smells, temperature, etc. *Theories used:* SEM, SCT and SILT to develop discussion part for caregivers to share their difficulties and understand that they are well-supported when they have problems while parenting. These theories were also used to develop games that allow children to relax and enjoy the calm nature brings in a peer group format. |
| 5 | **Environment and Fuzzy eating:** The environmental awareness and children, waste of food. Develop parental skills: how to feed/how to manage food rejection and demands. Connectedness to Nature/outdoor play: active nature games, discovering nature, practicing awareness to sounds, touch, smells, temperature, etc. *Theories used:* SEM, SDT, SCT and SILT to develop a peer-supported group discussion on fuzzy eating and activity to allow the children to touch, smell, and taste the raw vegetables before eating them with peers autonomously. |
| 6 | **Environment and Limit setting:** Provide caregivers with understanding about care for nature, responsibility and power struggle. Connectedness to Nature/outdoor play: active nature games, discovering nature, practicing awareness to sounds, touch, smells, temperature, etc. *Theories used:* SEM, SCT, SILT, THD to develop games for children to understand how nature grows, the essence of life and growth in nature, and eventually to learn to care for nature as a group with other peers. |
| 7 | **Environmental awareness and Fun with food:** Talking about resources and cooking together. To develop understanding about parental modelling, sedentary behaviors. Connectedness to Nature/outdoor play: active nature games, discovering nature, practicing awareness to sounds, touch, smells, temperature, etc. *Theories used:* SEM, SCT, SILT, THD to develop games and engagements for caregivers to "cook" together with their own children as well as other peers. "Cook" refers to preparing raw vegetables, which allows participants to appreciate the "patterns" or "drawings" inside of the vegetable when they are cut open. |
| 8 | **Environmental awareness and healthy habits:** Rules and routines. Develop skills on providing fail-safe food and activity environments and taking responsibility for nature. Throwing, catching, and bouncing skills. Connectedness to Nature/outdoor play: active nature games, discovering nature, practicing awareness to sounds, touch, smells, temperature, etc. *Theories used:* SEM, SCT, SILT, THD to let the caregivers know how important it is to set up rules and routines in the families, group discussion specifically increases the motivation for participants to develop healthy habits in the family. Games were designed to show how simple it is to incorporate active nature games in daily routines. |

**Table 3.** *Cont.*

| Session | Activities during the Program |
|---------|-------------------------------|
| 9 | **Nature and me:** Run & Fun: Promoting physical activity in Nature. Safety and fun in the nature. Develop parental skills for creating safe outdoor activities in nature environment. Connectedness to Nature/outdoor play: active nature games, discovering nature, practicing awareness to sounds, touch, smells, temperature, etc. *Theories used:* SEM, SCT and SILT to let the caregivers know how important it is to set up rules and routines in their family, group discussion specifically increases the motivation for participants to develop healthy habits in the family. Games were designed to show how simple it is to incorporate active nature games in daily routines. |
| 10 | **Farewell and graduation: Summary.** Connectedness to Nature/outdoor play: active nature games, discovering nature, practicing awareness to sounds, touch, smells, temperature, etc. *Theories used:* SEM, SDT, SCT, SILT, THD to set up a final group discussion to summarize all the topics and encourage participants to continue this nature-connected, healthy lifestyle routine. |

Families in the CG were asked to read the health-related dietary recommendations published by the Hong Kong government [41]; they believed that these government recommendations were part of the program. The participants of the CG were also offered the complete material package of *Play&Grow* after the follow-up assessment as an incentive.

*4.2. Participants*

Preschool children aged two to five, living in Hong Kong, were invited to participate in the *Play&Grow* program together with their main caregivers. They were recruited through advertisements at community centers, kindergarten schools, and both The University of Hong Kong's (HKU) and *Play&Grow*'s Facebook websites.

The exclusion criteria were children from non-local families and children with chronic health conditions. 639 (age 34.9 ± 9.5 months, 52.0% boys) families for the RCT were recruited in March 2017. The intervention retention rate was assessed by attendance sheets. Only children who attended at least 80% of the program sessions and completed their data were included in analyses (Intervention Group, IG, *n* = 467; Control Group, CG, *n* = 172) (Figure 2, Table 4). The initial allocation ratio was 2:1 with more dropouts in the Control. Participants who had any missing value in demographics at baseline or had a completion rate lower than 90% in any of the scales used in this study were excluded (e.g., declined to complete the program questionnaires or moved out of the country). Participants that dropped out did not differ in their demographic details or family socio-economic status when compared to the study population.

Information about child's gender, date of birth, family monthly income, parents' educational level, number of siblings, number of people in household, number of days child living with parents per week and child's primary caregiver in the daytime was collected at baseline on the first questionnaire. Age (months) was calculated with child's date of birth and date of questionnaire filled. Family monthly income was re-coded as "low to medium (≤HKD 40,000)" or "medium to high (>HKD 40,000). Father's and mother's educational levels were re-coded to "0" for "post-secondary and above" and "1" for "high school and below" according to original 4 categories (1 = "primary school", 2 = "secondary school", 3 = "high school", 4 = "post-secondary or above"). For the question of "only child or not", "0" referred to "no" and "1" referred to "yes". As for "child's primary caregiver in the daytime", "parents" (originally as "father" and "mother" on the questionnaire) was re-coded as "1" and "others" (originally as "grandparent", "domestic helper", and "others") was re-coded as "0".

**Table 4.** Demographic characteristics of the participants in both Intervention (IG) and Control (CG) groups. Survey Baseline.

| Demographics | | Participants before Random Assignment (*n* = 639) *n* (%) or *M* (*SD*) | IG (*n* = 467) *n* (%) or *M* (*SD*) | CG (*n* = 172) *n* (%) or *M* (*SD*) | *t*/χ² | *p* |
|---|---|---|---|---|---|---|
| **Children** | | | | | | |
| Age (month) | | 34.9 (9.5) | 34.7 (10.0) | 35.4 (8.2) | −0.844 | 0.40 |
| Sex | Boys | 332 (52.0) | 241 (51.6) | 91 (52.9) | 0.071 | 0.79 |
| **Families** | | | | | | |
| Primary Caregiver | Mother | 411 (64.3) | 299 (64.0) | 112 (65.1) | 6.093 | 0.30 |
| Mother's age (year) | | 35.5 (3.9) | 35.4 (3.86) | 35.74 (4.05) | −1.003 | 0.32 |
| Father's nationality | Chinese | 597 (93.4) | 438 (93.8) | 159 (92.4) | 1.753 | 0.42 |
| Mother's nationality | Chinese | 615 (96.2) | 448 (95.9) | 167 (97.1) | 0.46 | 0.98 |
| Family structure | Nuclear family | 465 (72.8) | 340 (72.8) | 125 (72.7) | 1.695 | 0.64 |
| Family monthly income | >40,000 HKD | 441 (69.0) | 308 (66.0) | 133 (77.3) | 8.867 | 0.11 |
| Father's education level | Post-secondary or above | 497 (77.8) | 365 (78.2) | 132 (76.7) | 0.354 | 0.84 |
| Mother's education level | Post-secondary or above | 552 (86.4) | 397 (85.0) | 155 (90.1) | 3.927 | 0.27 |
| Father's work situation | Employed | 530 (82.9) | 385 (82.4) | 145 (84.3) | 7.234 | 0.20 |
| Mother's work situation | Employed | 369 (57.7) | 249 (53.3) | 120 (69.8) | 15.591 | 0.004 |

HKD = Hong Kong Dollars; Independent group *t*-tests run for continuous variables; χ² tests run for categorical variables.

The IG and CG were similar in their demographics or family socio-economic status, apart from the mothers' employment (Table 4). The families were only informed of participating on a health program but not of the trial's hypotheses. All the questionnaires were completed prior to and after the intervention by the primary caregiver, without any assistance related to the questionnaires, such as explanation or clarification, to minimize interference. The typists and the team performing analysis were blinded to the allocation, measurements, and assessments.

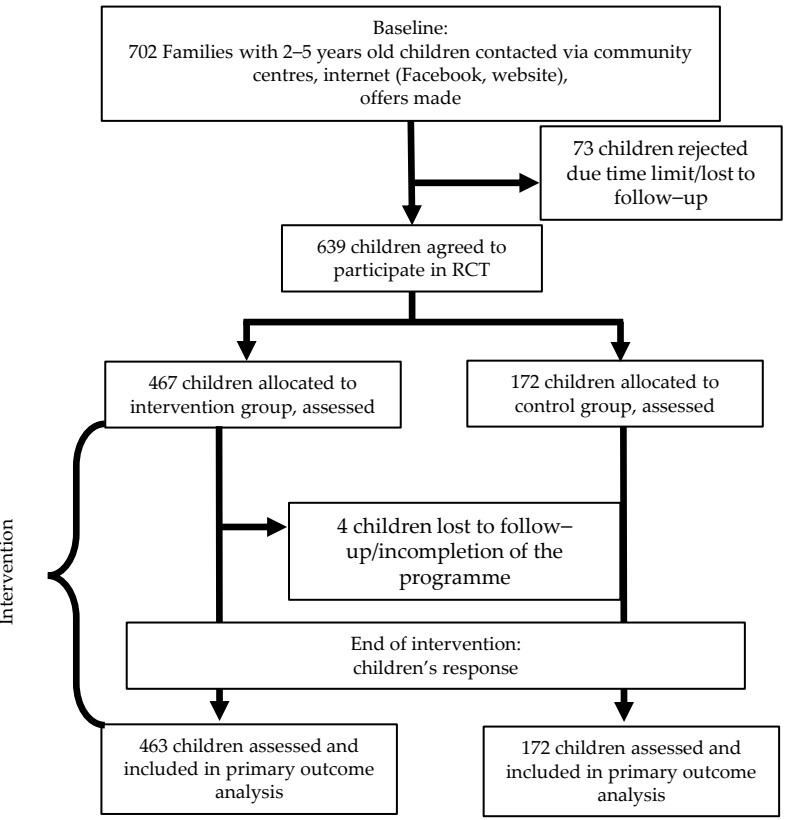

**Figure 2.** Timeline flow diagram.

*4.3. Outcome Measures*

4.3.1. Children's Connectedness to Nature

The Connectedness to Nature Index for Parents of Preschool Children (CNI-PPC) [3] was used to measure children's connectedness to nature. Parents rate their children's closeness to natural environments in the past three months for 16 items on a 5-point scale from 1 ("disagree") to 5 ("agree"). The inventory consists of 4 factors: enjoyment of nature (ENN, 6 items), empathy for nature (EMN, 3 items), responsibility toward nature (RN, 3 items), and awareness of nature (AN, 4 items). The total score and sub-scores are calculated by adding up the item scores, so the total score ranges from 16 to 80, with higher score indicating more connectedness to nature. The scale has been reported to have good reliability and validity for Hong Kong young children [3]. The Cronbach's α scores for the current sample were 0.89 for before and 0.90 for after intervention.

4.3.2. Children's Psychological Development

The 25-item Strengths and Difficulties Questionnaire (SDQ) for 2–4 year olds [19] was used to evaluate participants' psychological development conditions in the past three months. The SDQ consists of 5 factors: 4 difficulty factors (emotional problems, conduct problems, hyperactivity, and peer problems) and 1 strength factor (prosocial); each of these factors includes 5 items. The responses were rated on a 3-point scale with 0 as "not true", 1 as "somewhat true", and 2 as "certainly true". After reverse scoring for certain items, the sum for each subscale ranges from 0 to 10. Higher scores in difficulty subscales indicate higher levels of problems, whereas higher scores on the prosocial subscale indicates better prosocial behaviors. These scores in each subscale (except for the prosocial behavior scale) can be summarized into a total difficulties score that reflects emotional and behavioral difficulties. The parent reported SDQ for ages 2–4 has been demonstrated to have acceptable reliability and validity for Hong Kong preschoolers [3]. The Cronbach's α scores in this

study were 0.74 for before and 0.76 for after intervention, indicating good psychometric properties.

### 4.4. Statistical Analysis

All analysis began after the data collection was finished without any interim analyses in the study. The investigation on whether the SDQ changes were related to the CNI-PPC changes employed a repeated measures structural equation model with intervention status as a causal predictor. As previously described [3], the four factors of the CN were created theoretically and replicated in this study. The hypothesized relationship of CN to SDQ was exploratory but under the presumption that higher scores for CN would reduce difficulties and increase strengths in the SDQ as previously reported [3]. Significance was reported from the SEM software; SEM identifies the unique contribution of each factor to dependent variables, taking into account both the inter-correlated nature of the CN predictors and sample size. Path values that had $z$-scores with $p > 0.05$ were treated as not significant. Removal of non-significant paths improves model fit as such action indicates that the path is equivalent to zero in the data. This is a data-dependent procedure and explains why this study is important because this new data set allows us to test in an independent group the observed relationships in study [3] and to test them for equivalence before and after the intervention in control and experimental groups.

Factor and path analyses were conducted using a combination of the 'lavaan' package [42] in Jamovi v1.01 [43] and Amos [44]. Cases with more than 10% missing data in any factor were deleted, while cases with <10% missing data were imputed using the expectation maximization procedure [45]. The distribution of data was completely at random according to the ratio of Little's MCAR $\chi^2$ to $df$ ratio. Further details are available [18]. The sample size relative to the number of items was relatively low (4.10 cases per variable in the CG), so CN and SDQ scales were parceled into single item indicators and treated as indicators of the overall construct. To examine the effect of the intervention and to identify any causal effects, differences in paths' weights were examined.

To account for the hypothesized causal influence of the intervention on the children's PW, the repeated measures path model (Figure 1) had:

a. At both times, configural equivalent latent factor structures for CN (four scale score indicators) and SDQ (five scale score indicators);
b. Correlated residuals or factors from Time 1 to Time 2 to capture the correlated nature of constructs;
c. Statistically significant causal paths from experimental status as a dummy variable (0 = CG, 1 = IG) onto the CN factors at Time 2; and
d. Statistically significant regression paths from CN scale scores to SDQ scale scores at each time point.

Effect sizes for the $R^2$ variance explained were calculated as $f^2$, with values >0.15 interpreted as medium, and >0.35 as large. Once plausible scale scores were identified, the impact of the intervention was evaluated on an intention-to-treat basis. Multiple group confirmatory factor analysis with invariance testing was used to determine the extent of equivalence between EG and CG in the structural model. Differences in parameter values were evaluated by POL to ascertain those that were statistically significant [22].

### 4.5. Approval by Ethical Committee and Consent to Participate

The RCT was approved by the University of Hong Kong Human Research Ethics Committee for research involving human participants (EA1502073). It was carried out according to The Code of Ethics of the World Medical Association (Declaration of Helsinki) for experiments involving humans. The protocol of the RCT has been published [19,21]. Written informed consent was obtained from all caregivers prior to the study.

## 5. Conclusions

The new findings of this study were possible due to the systematic modeling of repeated measures over time and contrasts between IG and CG participants. This allowed us to observe the impact of the *Play&Grow* intervention on specific aspects of both CN and SDG and how those changes strengthen the relationship of CN responsibility for nature to prosocial strengths. This study provided empirical evidence that connecting children with nature was beneficial to their mental development. Specifically, the increase in children's Responsibility for Nature showed the strongest impact, relating to improvement in hyperactivity and prosocial behaviors. A recent report on mental health in preschoolers suggested screening and early identification of children with psychological problems and their enrollment into evidence-based interventions [1,46,47] as the children with highest risk respond well on the early lifestyle interventions [46,47]. *Play&Grow* can be one such intervention as well. The CN tool presented in this study is ready for broader exposure, especially in research. The next steps should include (a) further validation of this tool in various contexts and populations and (b) further investigation, based on other prominent studies mentioned in this paper and as well as on this one, of the influence of connectedness to nature on psychological well-being.

**Author Contributions:** T.S. was the principal investigator of this trial, designed the trials and the study protocol, and drafted and revised the manuscript. G.T.L.B. was responsible for data analysis and its interpretation. T.S. demonstrated all sessions with the help of trained instructors. T.S. and G.T.L.B. contributed to the literature search, data interpretation, and drafting of the manuscript. Both authors have read, contributed to, and approved the final version of the manuscript. T.S. had final responsibility for the decision to submit for publication. Both authors have read and agreed to the published version of the manuscript.

**Funding:** The study is supported by HKU SEED funding and General Research Grant, HKSAR, nr 17108217. The funder (the University of Hong Kong) had no role in the study's design, data collection, analysis, interpretation, or the writing of the article.

**Data Availability Statement:** The dataset is available on https://doi.org/10.17608/k6.auckland.14 552652.v1.

**Acknowledgments:** The authors would like to thank the *Play&Grow* staff members, and the families who participated in the program.

**Conflicts of Interest:** The authors declare no conflict of interest. The funders had no role in the design of the study; in the collection, analyses, or interpretation of data; in the writing of the manuscript, or in the decision to publish the results.

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
