# Peer review of "The Influence of Connectedness to Nature on Psychological Well-Being: Evidence from the Randomized Controlled Trial Play&Grow"

_challenges, doi:10.3390/challe12010012_

Round 1

Reviewer 1 Report

This is a timely study, especially as children come out of restrictions caused by the COVID-19 pandemic.  The authors carried out a randomized control study to find out whether connectedness to nature would improve the psychological well-being of pre-school children, using the interventions of the Play&Grow scheme, into which they introduced a new element - connectedness to nature.  The children's psychological well-being was measured by the Strengths and Difficulties (SDQ) checklist. The results are very encouraging. The experimental group showed a stronger sense of responsibility to nature and lower levels of difficulties in the SDQ  hyperactivity scales; they had fewer peer problems and higher prosocial scores. Emotional improvement levels were positively related to increasing awareness of nature. The authors provide a credible explanation of why these factors might be linked. They are also ready to acknowledge that results might differ in different cultural contexts.  This study was carried out in Hong Kong and it would be interesting to replicate it in different social and cultural contexts.  The activities in the Play&Grow scheme look very child-friendly and could easily be adapted.  It is to be hoped that this kind of intervention is implemented and evaluated in many countries.

Author Response

We appreciate the reviewer positive view on this manuscript, acknowledgement of the critical importance to further implement and evaluate this kind of intervention in many countries.

Reviewer 2 Report

This is an interesting study highlighting the integrated and interactive nature of socio-emotional-psychological wellbeing and environmental influences. The fact that environmental conditions/circumstances can be manipulated/adjusted and tested and their impact can be evaluated using "soft" (standardized social/behavioral measures/indicators)  and "hard" (physiological measures/indicators) as is the case in this study, is intriguing. 

My  recommendation is that the authors present the results, discussion, and conclusion sections after presenting the methodology of the study (section 4).

Author Response

We sincerely thank the reviewer for this review, especially noting the use of "soft" (standardized social/behavioral measures/indicators)  and "hard" (physiological measures/indicators).

When it comes to the recommendation, we agree with the logic to use the methods before the results, but simply followed this journal's template and instructions for authors, which is the following:

Research manuscript sections: Introduction, Results, Discussion, Materials and Methods, Conclusions.

Reviewer 3 Report

The experimental study reported upon here is a very strong one, and methods are generally very well thought out.  Establishing associations between connectedness to nature and psychological development in young children would certainly represent an important insight.

My biggest concern is that there are clearly a number of previous papers on the same study, and the position of the current paper in relation is not made particularly clear.  The manuscript repeatedly references the other papers, but it also speaks in terms of having designed and conducted the study as if it were for the purposes of the current paper.  The authors should be clear to delineate that this is a secondary analysis – or so it seems, as the protocol paper certainly says nothing about SEM models – and to indicate how it adds to the other work, and what the motivations for this additional analysis were. 

Other comments primarily concern what I would say is overinterpretation of many results.  I feel as if a bit too much is being inferred from the single model.  Specific comments follow.

  • Abstract: SDQ is never actually defined, before the Methods.
  • Abstract: “The study showed that the impact shifted from the total CN score to the specific CN 20 factors.”  Not sure what this even means, here.
  • Abstract: It seems like a stretch to refer to the SDQ as “physiological.”
  • Intro par 1: watch grammar.  “changes in the how.”  “and the stability of how”
  • 53-65: I feel you are conflating two things here.  That CN enhances environmental sentiment is easy to believe.  But that it increases empathy or prosociality is harder to show, and doesn’t seem to follow from the former statement.  It just feels like you’re stretching results one step beyond what can be supported.
  • 86-88: I really don’t see how this conclusion follows from the stated statistics.  The correlations might be slightly higher for one concept that for the other; but do you have some sort of statistics to support the comparison?
  • 94-96: This reasoning for the use of SEM vs total scores might be presented earlier.  I was wondering about that all along.
  • 114-122: Although I trust the computed coefficients, the results seem to be either misinterpreted or at least overinterpreted.  In the first place, although the coefficients differ somewhat between pre and post time points, there is no indication that you tested for significance of the difference between these coefficients.  So how do you know they are really different?  Secondly, the paths from CNT to SDQT are presumably for all participants, across both groups.  So even if they are, say, bigger than pre-intervention, how can that be assumed to be an intervention effect?  Did you check for some sort of interaction, to see if path weights differed between groups?
  • As far as I can tell, Table 1 doesn’t show any association between CNT and SDQ, as asserted. Also, what analysis generated the numbers in this table?
  • 146-155 and Table 3: This is fine, in a descriptive sense.  But how does it show any sort of intervention effect?  It seems as if you would want some sort of analysis of pre-to post intervention change, or of post-intervention scores adjusted for pre.  It also seems odd that as much emphasis is put on differences pre-treatment, that must be completely random.
  • As far as I can tell, Table 2 isn’t referenced in the text.
  • Table 4: Again, I’ll accept the coefficients.  But it’s just not clear to me whether differences are large, or significant, or what?
  • Section 3.2: Again here, I think you’re rather speculating about connections that are beyond anything the study can support.
  • 243-252: I think you’re confuting two things here.  That awareness of nature is associated with enjoyment of nature doesn't imply that you will have greater emotional health in general.
  • 302-305: Need to say what effect size you are powered to detect, and why that size was chosen.
  • 332-341. I’d be curious to know the original allocation ratio and dropout rate in each group.  Perhaps a 2:1 ratio, with more dropouts in Control?  This just isn’t clarified, even though some other info about dropouts is given.   In Figure 2, is there a period prior to enrolment/randomization, in which participants could be lost to follow-up?  I’m not sure I’m getting the sequence of events.
  • 373-374: It’s self-referential to cite your own paper on the same study, to claim reliability of the measure.
  • 387-388: Same thing. “It has been reported …”  But these are really stats from the same study on which you are reporting!
  • Statistical Analysis: Make clear how factor structure was determined.  Just a latent factor loading onto the pre-determined subscales?  Also, how was significance calculated, to determine which paths to include?  Letting significance in simple tests guide choice of included paths is somewhat dubious, and it should be justified.
  • 415-420: This doesn’t seem particularly representative of what was done in the paper.

Author Response

Dear Reviewer,

Please find attached

Kind regards,

TS and GB

Reviewer 4 Report

  1. part 4 on materilas and methods must be placed before results
  2. conclusions should be extended to include what are the next steps in researching  connectedness to nature on psychological well-being, based on other prominent studies and espcially based on this one

Author Response

We sincerely thank the reviewer for this positive review and suggestions

1. We agree with the logic to use the materials ad methods before the results, but we simply followed this journal's template and instructions for authors, which is the following:

Research manuscript sections: Introduction, Results, Discussion, Materials and Methods, Conclusions.

2. This is a relevant comment and we have addressed the issue in the manuscript, suggesting in the conclusions the next steps in this novel area of connectedness to nature and psychological well-being, based on other prominent studies and especially on this one.

- "ThCN tool presented in this study is ready for broader exposure, especially in research. The next steps should include a) further validation of this tool in various contexts and populations and b)further investigation, based on other prominent studies mentioned in this paper and as well as on this oneof the influence of connectedness to nature on psychological well-being"(lines 437-440) at the end of the Conclusions.